# Evaluation of a Pharmacist-Led Methicillin-Resistant *Staphylococcus aureus* Nasal PCR Testing Protocol

**DOI:** 10.3390/antibiotics13121195

**Published:** 2024-12-07

**Authors:** Blain Thayer, Jonathan D. Edwards, Madeline G. Belk, Spencer H. Durham

**Affiliations:** 1Missouri Health University Hospital, Columbia, MO 65212, USA; 2Huntsville Hospital, Huntsville, AL 35801, USA; 3Department of Pharmacy Practice, Auburn University Harrison College of Pharmacy, Auburn, AL 36849, USA

**Keywords:** methicillin-resistant *Staphylococcus aureus*, antimicrobial stewardship, vancomycin, pneumonia, polymerase chain reaction

## Abstract

**Background/Objectives:** Methicillin-resistant *Staphylococcus aureus* (MRSA) can cause cases of community-acquired pneumonia, hospital-acquired pneumonia, and ventilator-associated pneumonia, and nasal colonization with this pathogen increases the risk of infection. Due to its high negative predictive value, multiple studies support using the MRSA nasal polymerase chain reaction (PCR) test to discontinue antimicrobials that target MRSA in the setting of a negative test result. The purpose of this project was to assess the utility of a protocol to allow pharmacists the ability to order MRSA nasal PCR screenings in hospitalized patients with pneumonia. **Results**: The pre-protocol group included a random sample of 100 patients, and the post-protocol group included 625 patients. Vancomycin DOTs when pharmacists ordered PCRs were significantly lower compared to the pre-protocol group (*p* < 0.5; 95% CI, 0.46–2.24). The average length of stay and readmission rates at 30 days were significantly lower in the post-protocol group compared to the pre-protocol group (*p* < 0.05 and *p* = 0.02, respectively), but there was no significant difference in mortality (*p* = 0.33). **Methods**: A protocol was implemented at our institution that allowed pharmacists to order an MRSA nasal PCR test in patients with pneumonia. This retrospective chart review compared a cohort of patients who received vancomycin from before implementation of the protocol to patients who received vancomycin after the protocol’s implementation. The primary endpoint was vancomycin days of therapy (DOTs) between the pre-protocol group and the post-protocol group. Other endpoints assessed included the length of hospitalization, readmission rates, and mortality. **Conclusions**: Pharmacists ordering MRSA nasal PCR tests significantly reduced vancomycin DOTs, average length of stay, and 30-day readmission rates, contributing to positive outcomes in patients with pneumonia.

## 1. Introduction

Community-acquired pneumonia (CAP), hospital-acquired pneumonia (HAP), and ventilator-associated pneumonia (VAP) are significant causes of morbidity and mortality in hospitalized patients. Methicillin-resistant *Staphylococcus aureus* (MRSA) is associated with all three types of pneumonia, though to varying degrees. Historically, MRSA was a rare cause in CAP and generally followed influenza infection, but it has slowly been increasing in incidence, with one study demonstrating an incidence of 8.9% (Rubenstein 2008). Conversely, it is a much more common cause of HAP and VAP, with incidences of greater than 20% in HAP and up to 15% in VAP [1,2,3]. MRSA is a well-known colonizer of the nares, which is a risk factor for subsequent infection [4,5,6]. Nasal screening for MRSA colonization has historically been used as an infection prevention measure to reduce transmission to others by identifying patients for contact isolation and potential decolonization [7,8,9]. In recent years, multiple studies support the use of MRSA nasal screening as an antimicrobial stewardship measure to de-escalate empiric MRSA antimicrobial coverage in the absence of MRSA nasal colonization [10,11,12,13,14,15]. These studies support that the MRSA nasal polymerase chain reaction (PCR) test has an extremely high negative predictive value (NPV) for MRSA pneumonia, usually greater than 95%, when the PCR test is negative, but it has a poor positive predictive value, indicating that antibiotics active against MRSA can be discontinued in the setting of a negative MRSA nasal PCR. The clinical practice guidelines from the American Thoracic Society and the Infectious Diseases Society of America for the treatment of CAP and HAP/VAP recommend the inclusion of antimicrobials with coverage against MRSA, such as vancomycin or linezolid, in patients with certain risk factors for MRSA infection, leading many patients to receive these antimicrobials empirically [16,17]. Therefore, a negative MRSA nasal PCR test, which can result in as little as 90 min, can quickly lead to the discontinuation of antimicrobials active against MRSA, making it a powerful tool for antimicrobial stewardship. Traditionally, prescribers, such as physicians or mid-level practitioners, have been responsible for ordering the MRSA nasal PCR. However, pharmacists play a pivotal role in the provision of antimicrobial stewardship, yet data are lacking on the ability of pharmacists to obtain MRSA nasal PCRs. The purpose of this project was to assess the utility of a protocol to allow pharmacists the ability to order MRSA nasal PCR screenings in hospitalized patients with pneumonia.

## 2. Results

A total of 625 patients were included in the post-protocol analysis. A comparison of the baseline demographics between the pre-protocol group and post-protocol group can be seen in Table 1. The groups were similar, with most patients being in their mid-60s, male, and white. In both groups, HAP was the most common type of pneumonia diagnosed.

Of the 625 patients in the post-protocol group, pharmacists ordered PCRs in 95 patients (23%), physicians in 319 patients (77%), and PCRs were not ordered in 211 patients (34%). Of the patients who had a PCR test, 82% were negative and 18% were positive. The vancomycin DOTs in both the pre-protocol and post-protocol groups are shown in Table 2.

The vancomycin DOTs when pharmacists ordered the PCRs were significantly lower compared to the pre-protocol group, when only prescribers ordered the PCRs (*p* < 0.5; 95% CI, 0.46–2.24), as shown in Figure 1.

Although the vancomycin DOTs were numerically lower when pharmacists ordered PCRs compared to when prescribers ordered PCRs in the post-protocol group, this was not statistically significant (*p* = 0.09; 95% CI, −1.15–0.09). The results of the secondary clinical outcomes can be seen in Table 3. The average length of stay and readmission rates at 30 days were significantly lower in the post-protocol group compared to the pre-protocol group, but there was no significant difference in mortality.

Approximately one-third of patients in the post-protocol group did not have an MRSA nasal PCR ordered, indicating that these were potential opportunities for pharmacist intervention. When pharmacists at the institution were surveyed on their level of confidence at ordering MRSA nasal PCRs, only 49% indicated they were confident, while 26% indicated they were either not comfortable or did not feel they had appropriate training to be able to order the test. Further, 21% of pharmacists either forgot or were unaware that they had the ability to order the test. The results of the pharmacist survey can be seen in Figure 2.

## 3. Discussion

To our knowledge, this is the first study to examine the effects of pharmacists having the ability to order MRSA nasal PCRs. The ability to de-escalate unnecessary antibiotics is a vital tool for antimicrobial stewardship, and pharmacists play a significant role in the provision of antimicrobial stewardship. In this study, pharmacists were given the ability to order MRSA nasal PCRs in patients diagnosed with pneumonia when the test had not already been ordered by the prescriber, usually a physician or nurse practitioner. Pharmacists can then contact the prescriber to discontinue vancomycin when the MRSA nasal PCR is negative, and there are no other potential sources for MRSA.

Vancomycin DOTs were significantly lower when pharmacists were able to order the test in the post-protocol group compared to the when only prescribers could order the test in the pre-protocol group. Interestingly, in the post-protocol group, the DOTs were numerically lower when pharmacists ordered the test compared to prescribers, but this was not significant. We hypothesize that this may have been due to a couple of different reasons. First, the pre-protocol group included only 100 randomly identified patients rather than all patients diagnosed with pneumonia during that study period, and the DOTs may have been more elevated in this group due to chance. Second, and perhaps more likely, after the pre-protocol period, prescribers and pharmacists were provided education on the utility of MRSA nasal PCRs in preparation for pharmacists being given the ability to order the test. This education may have resulted in prescribers ordering the test more frequently and discontinuing vancomycin when the patient had a negative test result in the post-protocol group compared to the pre-protocol group, making the vancomycin DOTs more comparable to when pharmacists ordered the test.

Despite the overall success of pharmacists ordering the test, one-third of patients who were eligible to receive an MRSA nasal PCR did not have the test ordered by a pharmacist or prescriber. This was an area of improvement identified for our institution as a whole, and particularly an area where pharmacists could have played a greater role. In the survey conducted of pharmacists to assess their level of comfort with ordering the test, only half responded with a positive level of comfort. Had more pharmacists responded positively, more PCR tests may have been ordered. In response, a mandatory education module on ordering the PCR test has been incorporated into training for newly hired pharmacists and is reiterated to the pharmacy staff at periodic meetings throughout each year. Lastly, the clinical outcomes of the average length of stay and readmission within 30 days were both significantly improved in the post-protocol group compared to the pre-protocol group. Although pharmacists contributed to these results, prescribers were included in these secondary outcomes as well. These results reinforce the clinical usefulness of the MRSA nasal PCR in the management of patients with pneumonia, regardless of which healthcare provider orders the test.

This study has important limitations that must be considered. First, it is a single-center, retrospective, observational analysis with limitations inherent to this type of study design. The pre-protocol group was a random sample of 100 patients and was completed for a different project to examine the use of vancomycin for the treatment of HAP and VAP only, whereas the post-protocol group included all patients diagnosed with pneumonia and receiving vancomycin, leading to a much higher number of patients in the post-protocol group. Additionally, the protocol which allows pharmacists to order MRSA nasal PCR tests allows them to be ordered in any patient diagnosed with pneumonia, including those prescribed linezolid instead of vancomycin for treatment. However, this project did not include an assessment of patients on linezolid, which limited a full assessment of the protocol.

## 4. Materials and Methods

This project was approved as Exempt by the Institutional Review Committee. Huntsville Hospital is a 971-bed community teaching hospital in Huntsville, Alabama, that services north Alabama and southern Tennessee. This project was originally conceived after conducting a medication use evaluation (MUE) of vancomycin for the treatment of HAP and VAP (henceforth known as the “pre-protocol” group). This MUE evaluated a random sample of 100 adult patients diagnosed with HAP or VAP and received vancomycin between June 16, 2021 and September 28, 2021. MRSA nasal PCR testing by prescribers only occurred in 37 of these 100 patients but was associated with a 100% negative predicted value for MRSA pneumonia in the 18 patients who also had sputum cultures performed. Based on the results of this MUE, combined with what was already known about the utility of MRSA nasal screenings, our institution concluded that MRSA nasal PCR screenings should be used more frequently to rule out MRSA and subsequently de-escalate therapy for HAP and VAP. In order to help increase MRSA nasal PCR screenings, a protocol was created to allow pharmacists the ability to order the test when an order had not already been placed (see Appendix A).

This retrospective chart review included all adult patients aged 18 years and older who received vancomycin for the treatment of CAP, HAP, VAP, aspiration pneumonia, or thoracic empyema between March 1, 2022 and August 31, 2022 after the implementation of the pharmacist-driven MRSA nasal PCR testing protocol (henceforth known as the “post-protocol” group). The primary endpoint was days of therapy (DOTs) for vancomycin between the pre-protocol group and the post-protocol group. Other endpoints assessed included length of hospitalization, readmission rates, and mortality. Descriptive statistics were used to characterize the sample, and Chi-square tests and Student’s *t*-test were used for the nominal data and continuous data, respectively. Additionally, a survey to assess pharmacist confidence level with ordering MRSA nasal PCRs was conducted.

## 5. Conclusions

In this study, the ability of pharmacists to order MRSA nasal PCR tests resulted in more tests being performed and a reduction in vancomycin DOTs. Additionally, both average length of stay and 30-day readmissions were significantly reduced, but there was no effect on mortality. Overall, these results can positively contribute to appropriate antimicrobial stewardship and improved outcomes in patients with pneumonia. As this is the first study to examine the ability of pharmacists to order this test, these results should be confirmed in larger and multisite clinical studies.

## Figures and Tables

**Figure 1 antibiotics-13-01195-f001:**
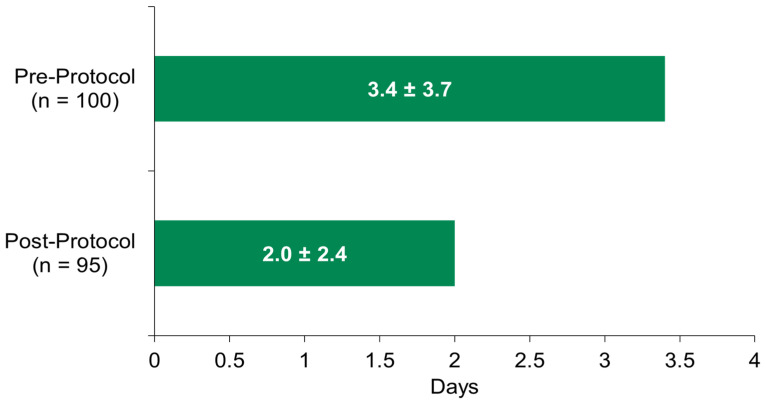
Vancomycin days of therapy—pre-protocol vs. post-protocol pharmacist-ordered PCRs.

**Figure 2 antibiotics-13-01195-f002:**
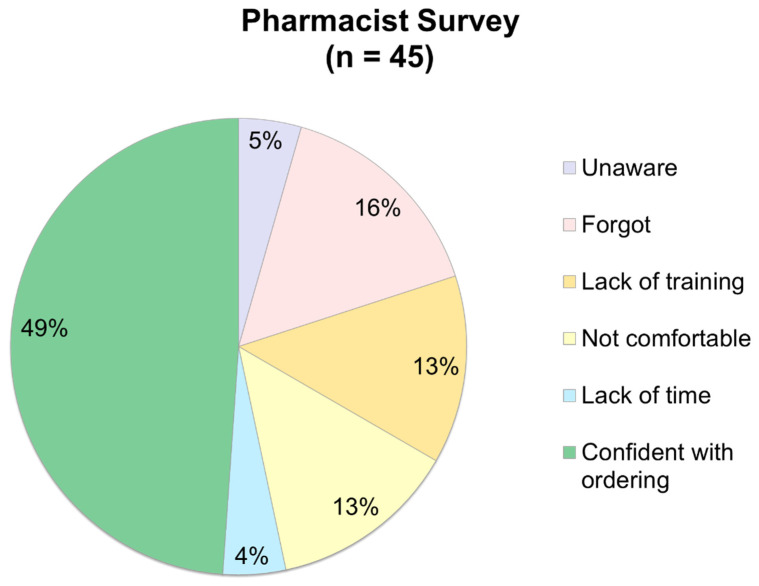
Pharmacist survey results.

**Table 1 antibiotics-13-01195-t001:** Baseline demographics in the pre-protocol and post-protocol groups.

Category	Pre-Protocol (n = 100)	Post-Protocol (n = 625)
Average age—years	66	67
Median age (IQR)—years	68	69
Male gender n (%)	58 (58%)	337 (54%)
White race n (%)	84 (84%)	490 (79%)
African American race n (%)	10 (10%)	115 (18%)
Other race n (%)	6 (6%)	20 (3%)
Hospital-acquired	97 (97%)	301 (48.2%)
Community-acquired	-	200 (32%)
Aspiration	-	104 (16.6%)
Ventilator-associated	3 (3%)	10 (1.6%)
Empyema	-	10 (1.6%)

**Table 2 antibiotics-13-01195-t002:** Vancomycin days of therapy in the pre-protocol and post-protocol groups.

Average days of vancomycin, pre-protocol	3.4 ± 3.7
Average days of vancomycin, post-protocol for all patients	2.0 ± 3.0
Average days of vancomycin, post-protocol for patients without PCR testing	3.0 ± 4.7
Average days of vancomycin, post-protocol for pharmacists-ordered PCRs	2.0 ± 2.4
Average days of vancomycin, post-protocol for physician-ordered PCRs	2.5 ± 2.8
Average days to discontinuation after negative PCR, post-protocol	1.9 ± 2.3

**Table 3 antibiotics-13-01195-t003:** Clinical outcomes.

Category	Pre-Protocol (n = 100)	Post-Protocol (n = 625)	*p*-Value
Average length of stay, days	13.2	11	< 0.05
Readmission within 30 days, n (%)	23 (23%)	87 (14%)	0.02
Death, n (%)	27 (27%)	138 (22%)	0.33

## Data Availability

The de-identified data presented in this study are available on request from the corresponding author due to privacy and HIPAA concerns related to the patients in this retrospective, observational study.

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
