# Peer review of "Evaluation of a Pharmacist-Led Methicillin-Resistant Staphylococcus aureus Nasal PCR Testing Protocol"

_antibiotics, 2024, doi:10.3390/antibiotics13121195_

Round 1
Reviewer 1 Report
Comments and Suggestions for Authors
I am grateful to you for assigning me a review of this brief report "Evaluation of a Pharmacist-Led Methicillin-Resistant Staphylococcus aureus Nasal PCR Testing Protocol" submitted by Blain Thayer and colleagues for publication in antibiotics.
In this study "Pharmacists ordering MRSA nasal PCR tests reduced vancomycin DOT by an average of 27% length of stay, and 30-day readmission rates, contribute to positive outcomes in patients with Pneumonia." This MRSA nasal PCR method is a powerful tool for antimicrobial stewardship.
Congratulations to the authors for this work.
Comments on the Quality of English LanguageThe quality of the English language is fine.
Author Response
Reviewer Comment:
I am grateful to you for assigning me a review of this brief report "Evaluation of a Pharmacist-Led Methicillin-Resistant Staphylococcus aureus Nasal PCR Testing Protocol" submitted by Blain Thayer and colleagues for publication in antibiotics. In this study "Pharmacists ordering MRSA nasal PCR tests reduced vancomycin DOT by an average of 27% length of stay, and 30-day readmission rates, contribute to positive outcomes in patients with Pneumonia." This MRSA nasal PCR method is a powerful tool for antimicrobial stewardship. Congratulations to the authors for this work.
Author Response: Thank you very much for your comments, they are much appreciated.
Reviewer 2 Report
Comments and Suggestions for Authors
The article is well-organized, and its novelty is appropriate for the field of antibiotics. I recommend publication without any additional comments.
Author Response
Reviewer Comment: The article is well-organized, and its novelty is appropriate for the field of antibiotics. I recommend publication without any additional comments.
Author Response: Thank you very much for your kind comments, they are very much appreciated.
Reviewer 3 Report
Comments and Suggestions for Authors
Methicillin-resistant Staphylococcus aureus (MRSA) infection is an important problem in both community- and hospital- acquired infections.
The authors of this manuscript evaluated the effect of MRSA nasal PCR tests led by pharmacists, and the tests actually contributed to the control of MRSA infections.
The results of this study can be useful in the control of MRSA infections.
I recommend that this manuscript is acceptable for publication in ANTIBIOTICS.
Author Response
Reviewer Comments: Methicillin-resistant Staphylococcus aureus (MRSA) infection is an important problem in both community- and hospital- acquired infections. The authors of this manuscript evaluated the effect of MRSA nasal PCR tests led by pharmacists, and the tests actually contributed to the control of MRSA infections. The results of this study can be useful in the control of MRSA infections. I recommend that this manuscript is acceptable for publication in ANTIBIOTICS.
Author Reponse: Thank you very much for your comments, they are much appreciated!